# An Evolving Hypergraph Convolutional Network for the Diagnosis of Alzheimer’s Disease

**DOI:** 10.3390/diagnostics12112632

**Published:** 2022-10-30

**Authors:** Xinlei Wang, Junchang Xin, Zhongyang Wang, Chuangang Li, Zhiqiong Wang

**Affiliations:** 1School of Computer Science and Engineering, Northeastern University, Shenyang 110169, China; 2Key Laboratory of Big Data Management and Analytics, Northeastern University, Shenyang 110169, China; 3College of Medicine and Biological Information Engineering, Northeastern University, Shenyang 110169, China

**Keywords:** Alzheimer’s disease, hyperbrain network, evolving hypergraph convolutional network, attention mechanism

## Abstract

In the diagnosis of Alzheimer’s Disease (AD), the brain network analysis method is often used. The traditional network can only reflect the pairwise association between two brain regions, but ignore the higher-order relationship between them. Therefore, a brain network construction method based on hypergraph, called hyperbrain network, is adopted. The brain network constructed by the conventional static hyperbrain network cannot reflect the dynamic changes in brain activity. Based on this, the construction of a dynamic hyperbrain network is proposed. In addition, graph convolutional networks also play a huge role in AD diagnosis. Therefore, an evolving hypergraph convolutional network for the dynamic hyperbrain network is proposed, and the attention mechanism is added to further enhance the ability of representation learning, and then it is used for the aided diagnosis of AD. The experimental results show that the proposed method can effectively improve the accuracy of AD diagnosis up to 99.09%, which is a 0.3 percent improvement over the best existing methods.

## 1. Introduction

Alzheimer’s disease (AD) is a classic neurodegenerative disease that results from weakened or lost functional connections in the brain. AD is often diagnosed clinically by functional magnetic resonance imaging (fMRI), which provides a non-invasive way to measure functional activities and changes in brains [1]. fMRI displays the activity of each brain region by measuring Blood Oxygen Level Dependent (BOLD) signals and discovers connectivities between brain regions. Brain networks can be constructed from such signals to represent functional connections between brain regions [2,3]. It is a simple description of the brain system. In brain networks, brain regions or neurons are usually defined as nodes, and the connection patterns between them are defined as edges. This portrayal of brain function provides a new tool for exploring the brain. In the analysis of brain networks, some patients with Alzheimer’s disease show abnormal patterns of connections in their brains, such as abnormal interruptions and changes between brain regions. Therefore, it is significant to study the diagnosis of AD through brain networks.

In recent years, there have been a large number of aided diagnosis methods using brain networks for brain diseases, such as the subgraph query method [4], kernel method [5,6], graph convolutional network method [7,8], etc. The traditional research on the diagnosis of AD is mostly carried out by extracting the topological features of brain functional networks [9,10]. Among these methods, the graph convolutional network method shows the best performance. It also takes into account the feature information and structure information of nodes. Compared with the traditional graph method, it can automatically extract features and learn more efficient features and patterns. Li et al. [11] propose a regional brain fusion–graph convolutional network (RBF-GCN), which is constructed with an RBF framework, mainly including three submodules, namely, the hemispheric network generation module, multichannel GCN module, and feature fusion module. In the multichannel GCN module, the improved GCN by our proposed adaptive native node attribute unit embeds within each channel independently. Ju et al. [12] use deep learning combined with brain networks and clinically relevant text information for early diagnosis of Alzheimer’s Disease. Song et al. [13] propose an auto-metric GNN (AMGNN) model for AD diagnosis. First, a metric-based meta-learning strategy is introduced to realize inductive learning for independent testing through multiple node classification tasks. In the meta-tasks, the small graphs help make the model insensitive to the sample size, thus improving the performance under small sample size conditions. Furthermore, an AMGNN layer with a probability constraint is designed to realize node similarity metric learning and effectively fuse multimodal data.

In recent years, the concept of hypergraph is put forward to express the high-order correlation between nodes [14]. According to this, the brain network based on hypergraph is also put forward, which is constructed by representing the correlation between one brain region and many other brain regions, so as to realize the related research of AD [15,16]. Different from the traditional brain network, the brain network constructed by hypergraph can depict the high-order interaction information of the brain region, which is called the hyperbrain network. Since the brain network itself is a kind of complex network, and existing studies have shown that a brain region usually only has a functional connection with several major brain regions [17,18], this characteristic just conforms to the construction mode of the hyperbrain network. In the hyperbrain network, edges represent interactions between multiple brain regions. At the same time, the neural network based on hypergraph has also received great attention in recent years, which can represent the high-order interaction information of brain regions, thus improving the classification performance. Xiao et al. [19] propose a hypergraph learning-based method, which constructs a hypergraph similarity matrix to represent the FCN, with hyperedges being generated by sparse regression and their weights being learned by hypergraph learning. The proposed method is capable of better capturing the relations among multiple brain regions than the traditional graph methods and the existing unweighted hypergraph methods. Ji et al. [20] propose a hypergraph attention network for functional brain network classification to further extract information in hypergraphs, which includes features among nodes and hyperedges.

In the construction of brain network, the method of calculating the correlation between brain regions is adopted. In fact, several studies have demonstrated that the FC of evolving brain networks exhibits dynamic changes in brain activities over time [21,22]. Zhang et al. [23] studied the statistical properties of complex networks with different time series and found that time series with different dynamics show different topologies. The research shows that the topology of the functionally connected network will change significantly with time. Therefore, it is necessary to consider this dynamic change in the diagnosis of AD. There have been some evolving graph methods for brain network classification in recent years. Cao et al. [24] propose a machine learning approach for the classification of neurological disorders while providing an interpretable framework. Firstly, an rs-fMRI time series is transformed into multigraph by using the sliding window technique. A temporal multigraph clustering is then designed to eliminate the inconsistency of the temporal multigraph series. Then, a graph-structure-aware LSTM is further proposed to capture the spatio-temporal embedding for temporal graphs. In recent years, the dynamic brain network has been proposed and used to reflect the dynamic changes in the brain signal information [25,26]. By dividing the signal of the brain regions by sliding windows, the correlation between each sliding window is calculated, and the sequence of the dynamic brain network is obtained.

According to this idea, a dynamic hyperbrain network is proposed. By dividing the signal of brain regions, the interaction between multiple brain regions is calculated for each segment of signal, thus constructing the dynamic hyperbrain network. Based on the dynamic hyperbrain network, the hypergraph convolutional network is realized, and the hypergraph attention network is further realized. This kind of evolving hypergraph convolutional network can not only represent the high-order interaction information of brain regions but also reflect the temporal variation information of the hyperbrain network, so as to improve the classification accuracy. Therefore, the evolving hypergraph convolutional network is adopted to diagnose AD.

The contributions of this paper are as follows:Inspired by neuroscience, the connection pattern of the brain is similar to the construction of a hypergraph, so the dynamic hyperbrain network is proposed for the diagnosis of AD.An evolving hypergraph convolutional network is proposed based on the hyperbrain network. Additionally, an evolving hypergraph attention network is further proposed to improve the accuracy of the diagnosis of AD.Experiments are performed with two kinds of evolving hypergraph convolutional networks on two real data sets, and the experimental results indicate that our proposed methods can effectively improve the accuracy.

## 2. Method

### 2.1. Dynamic Hyperbrain Network

First, the time series of brain regions is segmented into different time periods $t=\left\{t_{1},t_{2},\cdots ,t_{s}\right\}$ using a sliding window, then the hyperbrain network is constructed according to the signal from each time period.

In the traditional graph, an edge can only connect two nodes, so the traditional graph can only describe the pairwise relationship between a pair of nodes. In fact, in addition to pairwise relationships, there are higher-order relationships in many applications (such as functional interactions between multiple brain regions) that cannot be expressed by traditional diagrams. To solve this problem, a hypergraph [27] is proposed to depict higher-order relationships between multiple nodes. In the construction of super networks, sparse representation [28] is used, and the construction process is shown in Figure 1.

$Z^{t}={\left[z_{1}^{t},z_{2}^{t},\cdots ,z_{m}^{t},\cdots ,z_{M}^{t}\right]}^{T}\in R^{M\times d}$ represents the training data set including *M* brain regions, where $z_{m}^{t}$ represents the *t*th time series of the *m*th brain region, and *d* is the number of time points of the time series. Take each time series of brain region as a response vector, and the linear combination of the time series of the remaining M−1 brain regions is used to estimate this vector:(1)$$z_{m}^{t}=A_{m}^{t}\alpha_{m}^{t}+\varepsilon_{m}^{t}\left(m=1,2,\cdots ,M,t=t_{1},t_{2},\cdots ,t_{s}\right)$$
where *s* denotes the number of snapshots in the dynamic hyperbrain network, and $A_{m}^{t}=\left[z_{1}^{t},z_{2}^{t},\cdots ,z_{m-1}^{t},0,z_{m+1}^{t},\cdots ,z_{M}^{t}\right]$ represents a matrix, which includes all brain regions except the *m*th brain region. $\alpha_{m}^{t}$ is a weight vector, which measures the impact of other brain regions on the *m*th brain region. $\varepsilon_{m}^{t}\in R^{d}$ is a bias. In order to obtain the final representation of the hyperbrain network, the following optimization problem needs to be solved:(2)$$\min_{\alpha_{m}^{t}}{∥z_{m}^{t}-A_{m}^{t}\alpha_{m}^{t}∥}_{2}+\lambda {∥\alpha_{m}^{t}∥}_{0}$$

However, $l_{0}$ normal form is an NP problem. So the $l_{1}$ normal form is used instead of the $l_{0}$ normal form to obtain the approximate solution of the problem. Then, the optimization problem becomes the following form:(3)$$\min_{\alpha_{m}^{t}}{∥z_{m}^{t}-A_{m}^{t}\alpha_{m}^{t}∥}_{2}+\lambda {∥\alpha_{m}^{t}∥}_{1}$$
where λ>0 is the coefficient of regularization, which controls the sparsity of the hyperbrain network. The larger the value of λ, the sparser the hyperbrain network model. Then, the least-angle regression method is used to solve the $l_{1}$ normal form problem.

Finally, the hyperbrain network under each snapshot is built to obtain the dynamic hyperbrain network $z_{m}=\left\{z_{m}^{t_{1}},z_{m}^{t_{2}},\cdots ,z_{m}^{t_{s}}\right\}$. It can obtain the interaction information between one brain region and several other brain regions in different time periods.

### 2.2. Evolving Hypergraph Convolution

The hypergraph convolutional network applies the convolutional network commonly used for images in deep learning to hypergraph data and updates the node features by aggregating the neighbor features of the node. The evolving hypergraph convolution method not only needs to consider the node features of the current time but also needs to consider the node features of the evolving hypergraph and aggregate the neighbor node information of a node and its neighbor nodes information of the adjacent snapshot to update the node features, as shown in Figure 2.

The obtained evolving hypergraph can be expressed as $G=\left(G_{1},G_{2},\cdots ,G_{s}\right)$, whose *m*th snapshot is denoted as $G_{m}=\left(V_{m},E_{m}\right)$,where $V_{m}$ represents the set of nodes and $E_{m}$ represents the set of hyperedges. Each hyperedge ϵ∈E is assigned a positive weight $W_{\epsilon \epsilon }$. Similar to the adjacency matrix of a simple graph, the hypergraph G can be represented by an incidence matrix $H\in R^{N\times M}$, where *N* denotes the number of nodes and *M* represents the number of hyperedges. If the hyperedge ϵ is incident with a vertex $v_{i}$, $H_{i\epsilon }=1$, otherwise $H_{i\epsilon }=0$.

The hypergraph convolutional network carries out the embedding propagation of each node by measuring the transition probability between two nodes. Therefore, according to the definition of hypergraph convolutional networks, the evolving hypergraph convolutional network can be expressed as:(4)$$x_{i}^{\left(l+1\right)}=\sigma \left(\sum_{t=1}^{T}\sum_{j=1}^{N}\sum_{\epsilon =1}^{M}H_{i\varepsilon t}H_{j\varepsilon t}W_{\varepsilon \varepsilon t}x_{jt}^{\left(l\right)}P\right)$$
where $x_{jt}^{\left(l\right)}$ is the feature of the *j*th vertex in the *l*th layer. $\sigma \left(\cdot \right)$ denotes the non-linear activation function. $P\in R^{F^{\left(l\right)}\times F^{\left(l+1\right)}}$ is the weight matrix between the *l*th layer and the l+1th layer. The embedding information is optimized by gradient descent.

### 2.3. Evolving Hypergraph Attention

In order to further improve the accuracy of classification, an attention learning module that can express timing information is applied to H. The probabilistic model is proposed to measure the degree of connectivity. Consequently, it can describe the relationship between vertices more accurately. The evolving hypergraph attention is shown as Figure 3.

For a given node *j* and its associated hyperedge ϵ, the attention score in time period *t* can be expressed as:(5)$$H_{j\epsilon t}=\frac{\exp\left(e_{j\epsilon }\right)}{\sum_{k\in N_{j\left(t-1\right)}}\exp\left(e_{jk\left(t-1\right)}\right)+\sum_{k\in N_{jt}}\exp\left(e_{jkt}\right)+\sum_{k\in N_{j\left(t+1\right)}}\exp\left(e_{jk\left(t+1\right)}\right)}$$

In the equation $e_{j\epsilon }=\sigma \left(sim\left(x_{j}P,x_{\epsilon }P\right)\right)$, where $\sigma \left(\cdot \right)$ represents the non-linear function such as LeakyReLU and eLU, $N_{j}$ is the neighborhood of node *j*. $sim\left(\cdot \right)$ is the similarity function of each two nodes, which is denoted as
(6)$$sim\left(x_{j},x_{\epsilon }\right)=\alpha^{T}\left[x_{j}∥x_{\epsilon }\right]$$
where $\cdot ∥\cdot$ represents concatenation, and α is a weight vector used to output a scalar similarity value and the elements in α range from 0 to 1.

$H_{j\epsilon t}$ is brought into Equation (Equation 4) for the node update to realize the attention network of the evolving hypergraph, denoted as $x_{i}^{\left(l+1\right)}=\sigma \left(\sum_{t=1}^{T}\sum_{j=1}^{N}\sum_{\epsilon =1}^{M}H_{i\varepsilon t}H_{j\varepsilon t}W_{\varepsilon \varepsilon t}x_{jt}^{\left(l\right)}P\right)$, so as to complete the classification task.

## 3. Results

In order to prove the effectiveness of the two proposed evolving hypergraph convolutional networks, they are compared with traditional graph convolutional networks, hypergraph convolutional networks, and evolving graph convolutional networks. ADNI and OASIS, two commonly used and real collected databases, are used to classify normal people and AD patients and obtain their accuracy. The experiments and data processing are carried out on a computer with InterCorei73.4GHzCPU,8GB.

### 3.1. Experimental Settings

Data sets: The fMRI data are obtained from the OASIS (http://www.oasis-brains.org/, accessed on 10 December 2021) and ADNI (https://adni.loni.usc.edu/, accessed on 15 January 2022) public data sets, which contain neuroimaging datasets from normal control (NC) and AD groups. In the OASIS dataset, a total of 1000 samples are selected, and there are 700 cases in the training set and 300 cases in the testing set. In the ADNI dataset, there is a total of 214 samples. During the experiments, 150 samples are selected as the training set and 64 samples as the testing set. The static brain network, static hyperbrain network, and dynamic hyperbrain network are all generated from these fMRI images.

(1) The first set of experimental data is obtained from the OASIS dataset. A total of 1000 rs-fMRI subjects are obtained, including 516 patients with Alzheimer disease and 484 normal controls. Each subject collects 164 sequence images using a Siemens 3.0T scanner (SIEMENS AG, Berlin, Germany). The details of the data information are shown in Table 1.

(2) The second set of experimental data is obtained from the ADNI dataset. A total of 214 rs-fMRI subjects are obtained, including 107 patients with Alzheimer disease and 107 normal controls. There are 140 sequence images for each subject, and all images are acquired using a Philips 3.0T scanner (Philips Medical Systems, Andover, MA, USA), and other data information is shown in Table 2.

We implement the proposed evolving hypergraph convolutional network and hypergraph attention using Pytorch. In addition, we closely follow [29,30] in the process of the parameter setting and network structure. The first layer of the model consists of 8-dimensional hidden representation, and the second layer generates a 2-dimensional feature for classification of disease. The activation function of each layer is Exponential Linear Unit (ELU). $L_{2}$ regularization is applied to the parameters of network with λ=0.0003. Additionally, in hypergraph attention, dropout with a rate of 0.6 is applied to both inputs of each layer and the attention transition matrix. We employ a linear transform as the similarity function $sim\left(\cdot \right)$ in the calculation of the attention incidence matrix *H*, followed by LeakyReLU non-linearity with the negative input slope set to 0.2. We train the model by minimizing the cross-entropy loss and use the Adam optimizer with a learning rate of 0.005. In experiments, the 10-fold cross-validation is adopted, and the mean classification accuracy is calculated.

### 3.2. Analysis

#### 3.2.1. Baseline Method

In order to verify the advantages of the proposed evolving hypergraph convolutional network (EHyper-Conv) and evolving hypergraph attention network (EHyper-Atten), the existing static graph convolution network (GCN), static graph attention network (GAT), evolving graph convolutional network (Evo GCN), evolving graph attention (Evo GAT), static hypergraph convolution network (Hyper-Conv), and static hypergraph attention network (Hyper-Atten) methods are compared, and the experimental results are shown in Figure 4. It can be seen from the results that the hypergraph method can achieve better results than the graph method, and the attention mechanism is further improved than in the convolutional method, so the proposed dynamic hypergraph method has the highest classification accuracy. In the diagnosis of AD, the EHyper-Conv method can achieve 99.03% accuracy, and the EHyper-Atten method is slightly better than the EHyper-Conv method, which can achieve 99.09% accuracy.

#### 3.2.2. Parameters

In order to verify the influence of different parameters on diagnostic accuracy, we conduct experiments with different parameter values and discuss the experimental results.

**The number of snapshot *s*:** The time series is divided into different time periods in the process of construction of the dynamic hyperbrain network. Different snapshot numbers are used for the diagnosis of AD to explore their effect on the classification results. The experimental results are shown in Figure 5, which displays that the results are robust to *s*, which means that the accuracy changes slightly with the number of snapshots. Additionally, it peaks at s=5, and when the value of *s* is too large or too small, it will cause a decrease in accuracy.

**λ in Equation (Equation 3):** In the process of the construction of the dynamic hyperbrain network, a brain region can interact with several other brain regions, and a set of hyperedges is produced through the changes in λ for each brain region. Among them, a large value means that the network is sparser and the nodes contained in the hyperedges are fewer. Therefore, we use different values $\lambda =\left\{0.1,0.2,0.3,0.4,0.5,0.6,0.7,0.8,0.9\right\}$ to conduct experiments to explore their impact on the diagnosis results. The classification results are shown in Figure 6. It can be seen that the accuracy is improved gradually with the increase in λ value.

**Hidden representation:** We also discuss the influence of hidden representation in the network on classification results. As shown in Figure 7, classification accuracy increases with the increase in hidden representation, reaches the peak when the length is 16, and then tends to be stable. Moreover, the EHyper-Conv method is always better than the Hyper-Conv and GCN methods in various feature dimensions, which indicates that the graph structure of EHyper-Conv is more robust and has better capability in graph embedding.

## 4. Discussion

At present, the diagnosis of Alzheimer’s disease focuses on how to obtain effective features from the brain network. The brain network is a kind of graph that can describe the association between brain regions. The diagnosis of AD through brain network is now a research hotspot. The most traditional method is to extract topological features of the graph by brain networks and then perform machine learning based on these features to realize disease classification. However, this method only considers the clustering coefficient and other features of the graph, which has considerable limitations. Therefore, the classification method based on kernel function is proposed, and the topological features of graphs are further effectively utilized. However, with the development of deep learning in recent years, graph convolutional networks have been gradually applied to the analysis of brain networks. In the diagnosis of AD, the deep features of the brain network are extracted by convolutional operations, so as to improve the accuracy of diagnosis. On this basis, graph convolutional network based on the dynamic brain network is proposed for AD diagnosis. This method considers the dynamic change information of brain networks so as to obtain more node information. However, these methods are all based on brain network construction, which only considers the association between any two brain regions. Clinical studies have proved that the association between brain regions is the joint action of various brain regions, and only considering pairwise information is one-sided. Therefore, the research method based on a hypernetwork is proposed again. This hypernetwork can represent the association relationship between a node and all other nodes, represent the high-order relationship between nodes, and enrich the node information.

In order to fully obtain the brain network node features, we propose an evolving hypergraph convolutional network based on a dynamic hyperbrain network for AD diagnosis. This method not only can show the dynamic changes in brain activity information but also can reflect the higher-order interaction information between brain regions. According to the dynamic brain network model, combined with the graph convolutional method, the AD diagnosis method based on evolving hypergraph convolutional network is implemented.

In the construction of the dynamic hyperbrain network, the information of different time periods is obtained by time series segmentation, and then the hyperbrain network is constructed under each time period. Then, the hyperbrain network set of all time periods constitutes the dynamic hyperbrain network. The node information of the dynamic hyperbrain network is aggregated and transferred layer by layer, and the node representation of the dynamic hyperbrain network is obtained. Finally, the classification task of AD is realized through the pooling and classification layer.

This method can enrich node information and has a better ability in graph knowledge embedding. In addition, in order to study the important brain regions in the process of classification, we adopt the blocking method validation in some regions and connect the related brain regions to simulate the phenomenon of blocking of brain regions. It can be found that the superior frontal gyrus, cingulate gyrus, parahippocampal gyrus, middle temporal gyrus, and temporal pole play important roles in the process of classification, which is consistent with neuroscience research and has proved the effectiveness of our methods.

The accuracy of the AD diagnosis method based on an evolving hypergraph convolutional network was greatly improved, but this method does not take into account the directionality of interactions between brain regions. Therefore, we plan to study the convolutional network based on dynamic directed hypergraphs in the future so as to realize the diagnosis of AD. In addition, a method based on graphs can fully show a correlation between brain regions. However, it ignores the information in fMRI images, so we also plan to add image information on the basis of graph information and combine convolutional neural networks with graph convolutional networks.

## 5. Conclusions

In this paper, we focus on the evolving hypergraph convolutional network for the diagnosis of AD for the first time. Firstly, a kind of dynamic hyperbrain network is constructed, which replaces the pairwise association with the association between one brain region and several other brain regions. Then, we propose an evolving hypergraph convolution method which takes into account both the dynamic changes in brain activity and the higher-order interaction information between brain regions. On the basis of a hypergraph convolution network, the node information of adjacent time slices is aggregated. In addition, an attention mechanism is added to the graph convolutional network to weight the features of neighboring nodes to achieve the goal of improving the accuracy. Finally, the real data from the ADNI and OASIS datasets are used to diagnose AD. The experimental results show that the proposed evolving hypergraph convolution can effectively improve the diagnostic accuracy, and the hypergraph attention can be further improved, thus verifying the effectiveness of the proposed method. In the future, we plan to extend other graph neural networks to the dynamic hypergraph model. Additionally, the convolutional network based on dynamically directed hypergraph is intended to be constructed to realize the diagnosis of AD. Finally, the information of fMRI images is planned to be combined with graph information to realize a convolutional network.

## Figures and Tables

**Figure 1 diagnostics-12-02632-f001:**
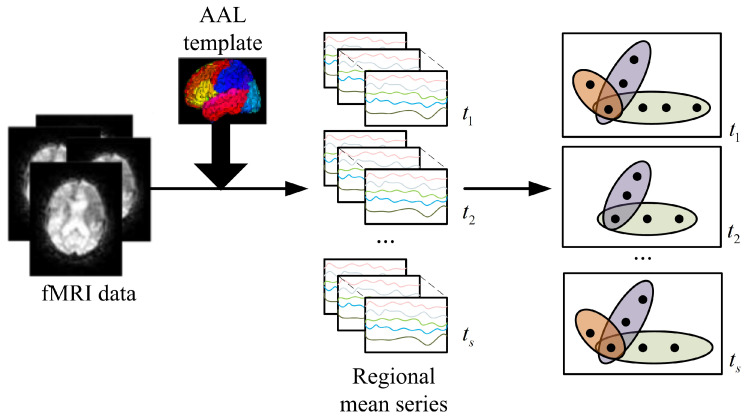
The construction of the dynamic hyperbrain network. (The fMRI image is matched with the AAL template to obtain the time-varying signal of each brain region. Then, the signal is divided into *s* time periods, and the sparse representation is calculated, respectively, to obtain the hyperbrain network of *s* time periods).

**Figure 2 diagnostics-12-02632-f002:**
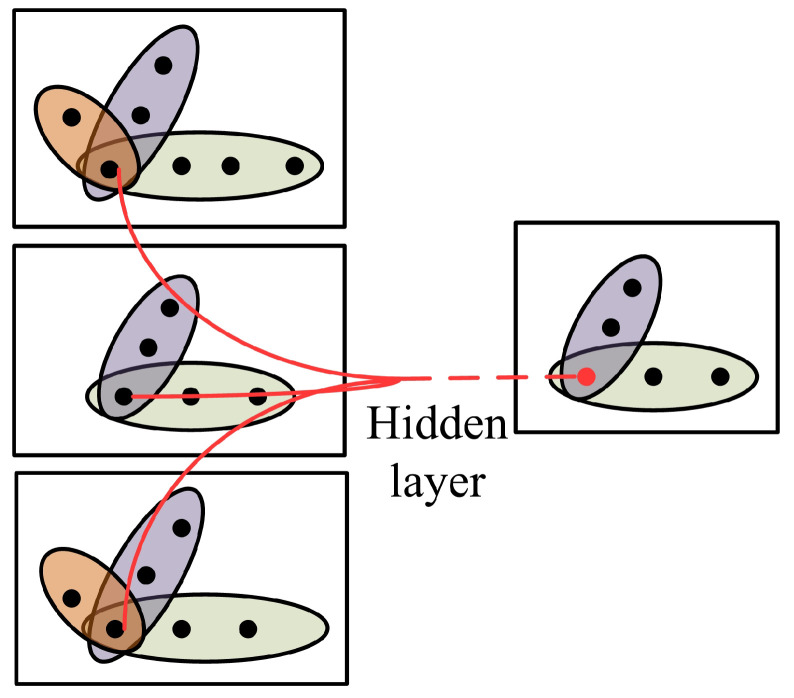
The embedding process of the evolving hypergraph convolution. (Node features in different time periods are aggregated and updated through the hidden layer.)

**Figure 3 diagnostics-12-02632-f003:**
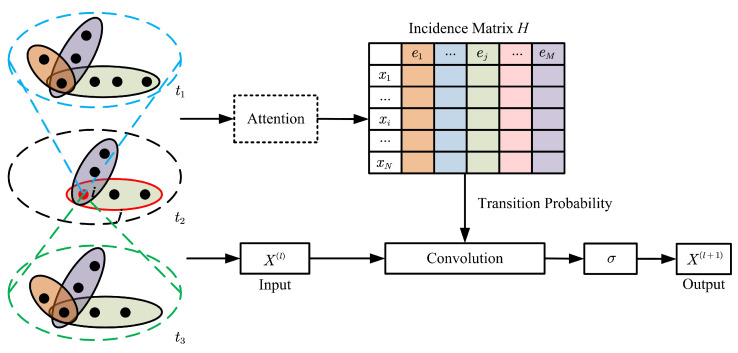
The process of evolving hypergraph attention. (Hypergraph convolution is upgraded to hypergraph attention by attention mechanism.)

**Figure 4 diagnostics-12-02632-f004:**
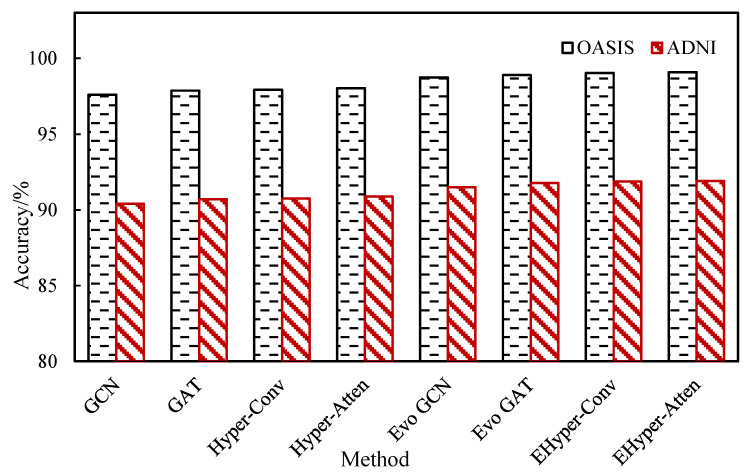
The comparison method in two data sets.

**Figure 5 diagnostics-12-02632-f005:**
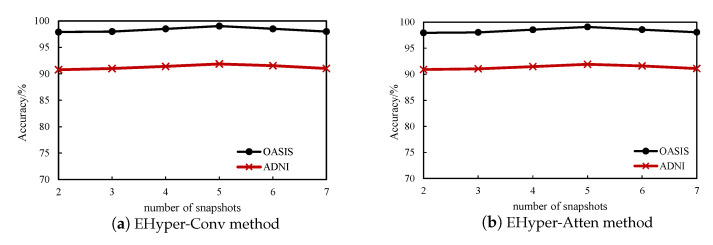
The effect of different number of snapshots in two methods.

**Figure 6 diagnostics-12-02632-f006:**
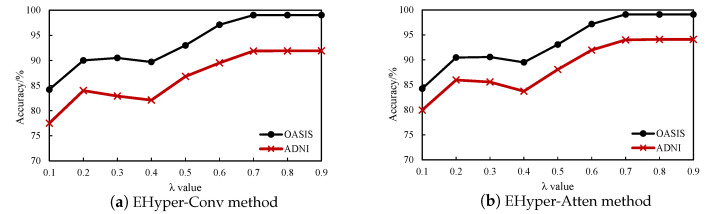
The effect of different λ values in two methods.

**Figure 7 diagnostics-12-02632-f007:**
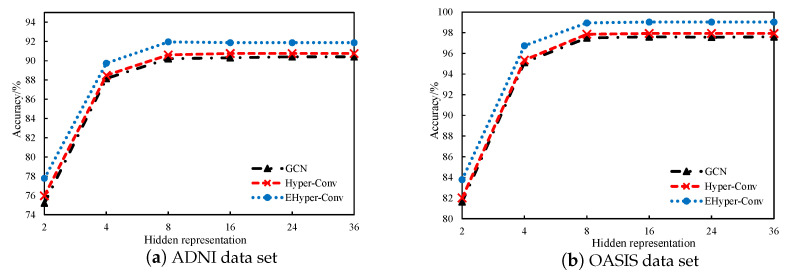
The effect of different hidden representations in two data sets.

**Table 1 diagnostics-12-02632-t001:** The details of OASIS dataset.

Type	Number of Subjects (Male/Female)	Age	Height/in	Weight/lbs
AD	269/247	77.49±16.44	64.5±9.5	182±66
NC	254/230	75.95±15.65	62.5±10.5	175±61

**Table 2 diagnostics-12-02632-t002:** The details of ADNI dataset.

Type	Number of Subjects (Male/Female)	Age	Weight/kg
AD	49/58	72±17	89.3±30.5
NC	53/54	70.5±15.5	80.9±28.3

## Data Availability

The data are from the Alzheimer’s Disease Neuroimaging Initiative (ADNI) and The Open Access Series of Imaging Studies (OASIS).

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
