# Peer review of "An Evolving Hypergraph Convolutional Network for the Diagnosis of Alzheimer’s Disease"

_diagnostics, 2022, doi:10.3390/diagnostics12112632_

Round 1

Reviewer 1 Report

The authors proposed a novel graph based deep learning approach for diagnosis of AD. The results look promising, but I have some doubts and hope authors could provide some clarifications.

1.     There are some typos and grammar errors in the writing. Please spend time to revise the manuscript.

2.     In Line 166, authors mention that sigma (.) represents the nonlinear function but where is sigma (.) in Equation (5)?  And where is sim(.) in this equation?

3.     What is the range for alpha in equation (6)?

4.     In Line 180, authors mention that the datasets are public, please provide references.

5.     The training, evaluation and testing details are missed. Please provide them.

6.     Please add some explanations on the legend of figures. For example, in Figure 1, what do these subfigures represent?

Reviewer 2 Report

This paper presents an interesting research work where the researchers proposed using a hypergraph convolutional network to diagnose Alzheimer’s disease. The article is written well and the research methodology followed is described well. I have the following comments before the paper can be accepted.

1- At the end of the abstract, you need to add some of the statistical results to give the readers a clear view of your results and the improvements that this research achieved in the field of Alzheimer's diagnosis.

2- No enough information about the used Dataset in the paper, you need to make a separate section for the used dataset that includes detailed information about it.

3- You need to add the parameter settings used for your experiments, not just mentioned your experiment setting like what is in [29] and [30].

4- No information about the machine specifications that was used to perform the experiments.

5- It is highly recommended to give some future research directions at the end of your conclusion.

Thank you.

Round 2

Reviewer 1 Report

The revision has addressed my comments and improved the quality of the manuscript 

Author Response

Thank you for your valuable comments.